# Antitumor Activity and Mechanism of Robustic Acid from *Dalbergia benthami* Prain via Computational Target Fishing

**DOI:** 10.3390/molecules25173919

**Published:** 2020-08-27

**Authors:** Juanjuan Huang, Ying Liang, Wenyu Tian, Jing Ma, Ling Huang, Benjie Li, Rui Chen, Dianpeng Li

**Affiliations:** 1School of Chemistry and Chemical Engineering, Guangxi University, Nanning 530004, China; hjjcbw@163.com; 2Faculty of Chinese Medicine Science, Guangxi University of Chinese Medicine, Nanning 530222, China; lbjjie15@163.com; 3School of Basic Medical Sciences, Guangxi Medical University, Nanning 530021, China; liangtty@163.com (Y.L.); bcyy15777199540@163.com (W.T.); huangling0916@hotmail.com (L.H.); 4School of Basic Medical Sciences, Guangxi University of Chinese Medicine, Nanning 530200, China; weixin0815@163.com; 5Guangxi Key Laboratory of Functional Phytochemicals Research and Utilization, Guangxi Institute of Botany, Guangxi Zhuang Autonomous Region and Chinese Academy of Sciences, Guilin 541006, China

**Keywords:** *Dalbergia benthami* prain, robustic acid, anti-tumor activity, computational target fishing

## Abstract

*Dalbergia benthami* Prain (*D.*
*benthami*) is an important legume species of the Dalbergia family, due to the use of its trunk and root heart in traditional Chinese medicine (TCM). In the present study, we reported the isolation, characterization and pharmacological activities of robustic acid (RA) from the ethyl acetate extract of *D. benthami* Prain. The SwissADME prediction showed that the RA satisfied the Lipinski’s rule of five (molecule weight (MW): 380.39 g/mol, lipid-water partition coefficient (log P): 3.72, hydrogen bond donors (Hdon): 1, hydrogen bond acceptors (Hacc): 6, rotatable bonds (Rbon): 3. Other chemical and pharmacological properties of this RA were also evaluated, including topological polar surface area (TPSA) = 78.13 Å and solubility (Log S) = −4.8. The probability values of the antineoplastic, anti-free radical activities and topoisomerase I (TopoI) inhibitory activity were found to be 0.784, 0.644 and 0.379, respectively. The molecular docking experiment using the Surflex-Dock showed that the Total Score and C Score of RNA binding with the human DNA-Topo I complex were 7.80 and 4. The MTS assay experiment showed that the inhibitory rates of RA on HL-60, MT4, Hela, HepG2, SK-OV-3 and MCF-7 cells were 37.37%, 97.41%, 81.22%, 34.4%, 32.68% and 51.4%, respectively. In addition, RA exhibited an inhibitory effect on the angiogenesis of zebrafish embryo, a good TopoI inhibitory activity at a 10 mM concentration and in a dose-dependent manner, excellent radical scavenging in the DPPH and ABTS assays, and the free radical scavenging rate was close to the positive control (BHT) at different concentrations (0.5–2.0 mg/mL). Furthermore, 18 potential targets were found for this RA by PharmMapper, including ANXA3, SRC, FGFR2, GSK3B, CSNK2B, YARS, LCK, EPHA2, MAPK14, RORA, CRABP2, PPP1CC, METAP2, MME, TTR, MET and KDR. The GO and KEGG pathway analysis revealed that the “protein tyrosine kinase activity”, “rap1 signaling pathway” and “PI3K-Akt signaling pathway” were significantly enriched by the RA target genes. Our results will provide new insights into the pharmaceutical use of this species. More importantly, our data will expand our understanding of the molecular mechanisms of RA functions.

## 1. Introduction

*D. benthami* is a legume species of the Dalbergia family. Its tree trunk and root heart are usually used in TCM. The main applications of *D. benthami* in TCM are in the therapy of traumatic injuries, as an analgesic and for the regulation of menstrual cycles [1]. It has been proven by modern pharmacology that *Dalbergia* genus plants have antitumor, antivirus and antioxidation potentials [2]. Furthermore, it has been shown that crude extracts of *D. benthami* have significant antibacterial and anti-inflammatory effects [3]. Previous studies of our lab showed that the ethyl acetate extract of *D. benthami* had a greater antioxidation than the extracts prepared with other solvents [4]. 14 compounds from *D. benthami* were isolated and identified as isoflavone, dalbergiben-thamone robustin acid, pseudobap-tigenin and formononetin, etc. [5]. Nevertheless, the active constituent of the ethyl acetate extract is unknown. In our previous studies, RA could be considered as one of the principal components in the plants from the *Dalbergia* genus, and it has been reported to be isolated in minute concentrations from other plants such as *Millettia thonningi* [6], *Erythrina indica* [7] and *Derris eriocarpa* How [8]. RA has also been synthesized in nine steps from methyl 2,4,6-trihydroxyphenyl ketone, with an overall yield of 18% [9]. The biological activities of RA were studied by the Chemical Genomics Center (NCGC), and it was found that RA was cytotoxic to H1299 lung cancer cells with p53 Mutation (IC50 7.94 µM) and that it also inhibits Cytochrome P450 activity (IC50 6.31 µM) [10,11]. RA also possesses a significant antimalarial activity [12]. In our previous studies, RA displayed an obvious Topo I inhibition activity and cytotoxicity, which was probably mediated by an arrest of the tumor cells at the G1 phase and a suppressed cell proliferation via the induction of apoptosis [13]. Nevertheless, the anti-tumor mechanism of RA still remains unclear, so the biological activities of RA need further exploration.

## 2. Results

### 2.1. The Chemical Structure, ADME Evaluation and Bioactivity Prediction of RA

We isolated a new pyranocoumarin from the ethyl acetate extract of *D. benthami* Prain, which was identified as RA (Figure 1a). It was shown that the structure of RA referred to a new linear pyranocoumarin. Next, we used the SwissADME software for an in-depth evaluation of the ADME-related properties of this RA. The values of Lipinski’s “rule of five”, such as the MW, log P, Hdon, Hacc and Rbon, can be seen in Table 1. In addition, the TPSA and solubility (Log S) of this RA were calculated as 78.13 (Å) and −4.80, respectively. Then, we predicted the biological activities of this RA using the PASS (Prediction of Activity Spectra for Substances) online tools and obtained the probabilities as being active (Pa) and inactive (Pi). We listed some of the most relevant activities according to the Pa values in Table 2 and found that the Pa values for the antineoplastic, anti-free radical activities and topo I inhibitory activity were 0.784, 0.655 and 0.379, respectively. To further analyze the topo I inhibitory activity of the RA by investigating the binding mode of the potent inhibitor of RA with the human DNA-TopoI complex, we carried out the molecular docking experiment using the Surflex-Dock in Sybyl 2.0 by Tripos Associates. The conformation was selected based on the Total Score (7.80) and C Score (4) for speculating the detailed binding mode in the site (Table 3). It showed that RA was located at the hydrophobic pocket, surrounded by the residues Glu356, Asn352, Asp533 and adenosine DC112, DA113 forming a stable hydrophobic binding. Furthermore, the oxygen atom in the amides group of the RA executed a moderate binding force by forming a hydrogen bond with the residues Arg362 (bond length: 2.83 Å) and Thr718 (bond length: 2.96 Å). Apart from the hydrogen bond force, the benzene ring of RA could also have an effect with the base pair DT10 and TGP11 in order to improve the π-π stacking force (Figure 1b).

### 2.2. Biological Activity Characterizations

We conducted the in vitro MTS assay to characterize the anti-cancer activities of this RA using some cancer cell lines. It showed that the half inhibitory activities of this RA in the HL−60, MT4, Hela, HepG2, SK-OV-3 and MCF-7 cells were 37.17%, 97.41%, 81.22%, 34.4%, 32.68% and 51.4% (Figure 2a), respectively. Figure 2b shows the conversion of supercoiled plasmid DNA by calf thymus Topo-I, RA and camptothecin (a well-known Topo I inhibitor, used as a positive control). It was revealed that the RA had the potential of binding to DNA fragments at a 10 mM concentration and that the exhibited inhibitory activities were concentration-dependent. Next, we performed an experiment to evaluate the anti-angiogenesis activity of the RNA in zebrafish embryos. Compared to the control, the length of subintestinal veins (SIVs) significantly decreased in the zebrafish embryos treated with this RA (Figure 2c,d). The results indicated an inhibitory effect of RA on the angiogenesis of zebrafish embryo. Furthermore, we evaluated the free radical scavenging activity of the RA. It was shown that this RA can serve as an effective scavenger against DPPH and ABTS radicals (Figure 2e,f). Its scavenging abilities increased steadily with the concentrations. All concentrations of RA had a scavenging effect on the DPPH and ABTS radical. RA possessed the highest scavenging capacity (74.85% and 82.5%) on the DPPH radical and ABTS radical at 2.0 mg/mL, respectively. Its scavenging rate was close to the positive control (BHT) at different concentrations.

### 2.3. In Silico Target Fishing for the RA

We employed the computational target fishing methods to predict the most probable target of this RA, and the workflow can be seen in Figure 3a. Using PharmMapper, we identified a total of 102 potential proteins (fit score > 3) that could be the targets of this RA, and using DARA-CPI we identified 125 potential targets (Z’-score < 0.5) for this RA. There were 18 targets shared by both methods (Table 4). Next, we performed the Gene Ontology (GO) and KEGG pathway analysis for these 18 targets. A total of 20 pathways (*p* < 0.05) were found to be involved by the targets of this RA (Figure 3b), including the “rap1 signaling pathway”, “PI3K-Akt signaling pathway”, “proteoglycans in cancer” and “focal adhesion”. We showed, in Figure 3c, the top 5 terms of the biological processes, cellular components and molecular functions from the GO annotation results (*p* < 0.01). Interestingly, “angiogenesis” and “transmembrane receptor protein tyrosine kinas signaling pathway” were found to be involved by the RA targets. The interaction network of this RA, its targets and potential pathways was visualized in Figure 3d. We found that 11 targets of this RA had roles in different pathways. 

## 3. Discussion

We reported and isolated a new pyranocoumarin, identified as RA, from the ethyl acetate extract of *D. benthami* Prain. The content of RA in the extract was estimated to be 3.5%. It has been reported that coumarins exhibit excellent anti-tumor, anti-inflammatory, anti-bacterial, antioxidant and other pharmacological activities [14]. Thus, our findings suggest that the RA can be considered as one of the principal components for therapy in the plants from the *Dalbergia* genus. The properties of this RA meet the requirements of Lipinski’s “rule of five”, and this suggested that it may be considered for compounds with oral delivery in mind [15]. Furthermore, the antineoplastic, anti-free radical activities and Topo I inhibitory activity of RA were predicted. It has been reported that the Topo I inhibitor activity could be the basis of anti-tumor activity [16,17]. In the previous report, we chose the conformation with the highest score (7.89) for analysis [13], and the differences with this manuscript were the hydrophobic residue and bond length. Nevertheless, they all indicated that the hydrophobic interactions between the benzene ring of RA and hydrophobic residues of the Topo I-DNA complex were likely to be responsible for the stability of the docked complex, RA inhibiting the Topo I activity in an allosteric manner. The biological activity experiments showed that RA displayed an inhibitory effect on tumor cells, Topo I inhibitory activity and anti-angiogenesis activity in zebrafish embryos, as well as on free radical scavenging activity, which was reported as the basis of anti-tumor activity. These results were consistent with the predicted results by PASS online.

Anti-angiogenesis might represent an important mechanism underlying anti-tumor activity. Oxidative stress represents a disturbance in the equilibrium status of prooxidant/antioxidant reactions in living organisms. The excess of reactive oxygen species (ROS) can damage cellular lipids, proteins or DNA. Oxidative imbalance has been implicated in a number of diseases, including cancers, atherosclerosis and heart diseases, as well as the ageing process [18]. The DPPH and ABTS free radical scavenging assays have been widely accepted for estimating the free radical scavenging activities of antioxidants. We showed that this RA possessed the highest scavenging capacity on the DPPH radical and ABTS radical at 2.0 mg/mL, at 74.85% and 82.5% (Figure 2e,f), respectively. 

We found SRC and SYK to be putative targets of the RA isolated in this study (Table 4). Some studies have shown a variety of malignancies characterized by the abnormal activation of SRC or SYK [19,20]. SRC and SYK are associated with cell survival, and their inhibition frequently leads to cell apoptosis, which is considered a promising therapeutic target for the treatment of malignancies [19,21]. We found that SRC and SYK were involved in the protein tyrosine kinase activity, kinase activity and protein kinase activity (Figure 3c). Tyrosine kinases are particularly important in the treatment of cancer and play a critical role in the development and progression of many cancers [22]. Over the last two decades, many molecules targeted by receptor tyrosine kinases (RTKs) have been used to treat a variety of tumors. RTKs mainly activate the PI3K/protein kinase B, rat sarcoma (RAS)/MAPK signal transducer and downstream effectors of multicellular processes during cancer progression [23]. We identified 18 targets of the RA that participated in 20 KEGG pathways (Figure 3b), some of which were involved in anti-tumor signaling pathways such as metabolic pathways, the PI3K/Akt signaling pathway and the NF-κB signaling pathway. These results indicated that the RA may exhibit anticarcinogenic effects. Furthermore, SYK is a regulator of mTOR and MAPK signaling in AML, and the inhibition of the PI3K pathway activity enhances the effects of SYK inhibition on AML cell viability and differentiation [24]. SRC overexpression acts synergistically on epithelial cell cloning and promotes tumorigenesis, depending on the JNK and PI3K signaling pathways [25]. Thus, we can speculate that RA may influence tumor progression by targeting the expression of PTK (SYK/SRC), inhibiting the activity of RAS and attenuating the PI3K/Akt pathway and NFκB pathway.

## 4. Materials and Methods

### 4.1. Sample and Reagents

The stems of *D. benthami* Prain were obtained from marketplace vendors from Guangxi Province, China, in June 2017. The voucher specimens were identified by Professor Songji Wei at the Department of Zhuang Pharmacy, Guangxi traditional Chinese Medical University, Nanning, China. 1,1-Diphenyl-2-picryl hydrazyl (DPPH) (purity 98%) was purchased from Wako Chemicals, California, CA, USA. 2,2′-Azobis-(3-ethylbenzothia zoline-6-sulfonic acid) (ABTS), camptothecin and Butylated hydroxyl toluene (BHT) were purchased from Sigma Aldrich Co. St. Louis, MO, USA. Other chemicals were purchased from China National Medicine Group Shanghai Corporation, Shanghai, China. All chemicals and solvents that were used were of analytical grade. 

### 4.2. Preparation of Plant Extract

The *D. benthami* Prain samples were initially air-dried and then powdered. The powdered plant material was passed through a 40-mesh screen prior to use. This powder was then suspended in 80% ethanol (1:10 drug to solvent ratio). The filtrate was collected three times at every four-day interval for a total collection period of 12 days. The extracts were concentrated, suspended in deionized water and sequentially partitioned with petroleum ether, ethyl acetate and *n*-butanol to obtain different fractions such as petroleum ether, ethyl acetate and n-butanol. The dry extracts were stored at −5 °C in the dark until further use.

### 4.3. The Isolation of RA

RA was obtained by repeated silica gel column chromatography from ethyl acetate extract in our group [26]. The compound was analyzed and confirmed as RA by GC-MS, ESI-MS, 1H-NMR and 13C-NMR to determine their chemical structures (Appendix A). The 1H-NMR data was found to be consistent with the literature values [10].

### 4.4. ADME Evaluation and Bioactivity Prediction

The ADME study was carried out using SwissADME (http://www.swissadme.ch). It is a free web-based tool to evaluate the pharmacokinetics, drug likeness and pharmacochemistry of small molecules [27,28]. The bioactivity prediction study was carried out using PASS (Prediction of Activity Spectra for Substances) online (http://www.pharmaexpert.ru/passonline/) software. PASS online tools has the ability to predict over 4000 kinds of biological activities and pharmacological effects, and was used in this study to check the bioactivity score of the compounds [29].

### 4.5. Bioactivity Study of Rustic Acid

#### 4.5.1. MTS Method

The in-vitro cytotoxicity test was carried out by MTS method. The SHL-60, MT4, Hela, HepG2, SK-OV-3 and MCF-7 cell lines were obtained from the Kunming Institute of Botany, Kunming, China. The cell suspension (100 μL) was cultured in a 96-well microplate and incubated at 37 °C for 24 h in a humidified 5% CO_2_ atmosphere. The medium was then replaced with solutions of RA (100 μg/mL). After the incubation period, 20 μL MTS Reagent (5 mg/mL) was added to each well, after which the plate was returned to the cell culture incubator for 4 h. The supernatants were removed, and 150 μL DMSO was added to all wells to initiate the colorimetric reaction. The absorbances were measured at 492 nm on an enzyme-linked immunosorbent assay microplate reader. Two positive compounds, cisplatin (DDP) and Taxol, were used in each experiment.

#### 4.5.2. Topo I Inhibitory Activity

The Topo I enzyme inhibitory activity was determined by measuring the conversion of supercoiled pBR322 DNA to relaxed isomers. Topo I and pBR322 were purchased from Takara Bio Inc., Tokyo, Japan. The reaction system (20 μL) was mixed with 2 μL reaction buffer solution, 0.5 μg pBR322 DNA, 2 μL of different compounds, 1 μL One unit of Topo I, and distilled water added to 20 μL, and this was reacted at 37 °C for 30 min. The reaction was terminated by adding 0.5% SDS, 0.25 μg/mL bromophenol blue and 15% glycerol. The reaction products were subjected to 0.8% agarose gel at 90 V for 50 min at room temperature, which was stained with 5 μg/mL ethidium bromide, after which imaging was conducted by the Bio-Rad Gel Doc**^TM^** XR+ system, Image Lab, California, CA, USA.

#### 4.5.3. Zebrafish Embryos Angiogenesis Assay

Healthy and limpid fluorescent embryos (German Tubingen strain, purchased from Model Animal Research Center of Nanjing University) were singled out at 24 h post-fertilization (hpf), digested with 1 mg/mL trypsin for 10 min at room temperature, and washed three times with PBS. Embryos were inoculated into a 96-well microplate (three embryos/well). Meanwhile, 5 μM or 10 µM RA was added to the culture media, and PBS served as the control. Embryos were maintained in a 28.5 °C incubator for an additional 48 h. 72 hpf embryos were plunged into 4% paraformaldehyde at room temperature for 2 h to be fixed, after which they were washed with PBS three times, dehydrated by soaking in 25, 50, 75 and 100% methanol in PBT buffer and gradually rehydrated to 100% PBT. The embryos were then balanced in NTMT buffer (0.1 M Tris-HCl at pH 9.5, 50 mM MgCl_2_, 0.1 M NaCl, 0.1% Tween 20) at room temperature for 0.5 h, and 375 mg NBT and 200 mg BCIP were added per mL NTMT buffer, respectively. After staining for 30 min, the reaction was stopped by adding PBST. The embryos were then plunged into a solution of 5% formamide and 10% hydrogen peroxide in PBS for 30 min. The photos were taken by fluorescence stereomicroscope (Olympus, Tokyo, Japan). Image J software was used to quantify the total length of SIVs. The antiangiogenic effect was defined as a reduction in SIVs’ length.

#### 4.5.4. Radical-Scavenging Activity

##### DPPH Radical-Scavenging Activity

The DPPH radical-scavenging activity of different fractions was measured with a modified method, described by Shimada [30]. In brief, RA solutions (100 μL) in 95% ethanol at different concentrations (0.5–2.0 mg/mL) were added to a 4 mL 0.004% (*w*/*v*) solution of DPPH in 95% ethanol. The reaction mixtures were incubated at 28 °C. The scavenging activities on the DPPH radical were determined by measuring the absorbance at 515 nm after 10 min. The antioxidant activity was expressed as a percentage of the scavenging of DPPH: SC% = [1 − (absorbance of sample)/(absorbance of control)] × 100%. The control contained all reagents, except the extract. The DPPH radical scavenging activities of BHT at the same concentration were also assayed for comparison. All tests were performed in triplicate, and the means were centered.

##### ABTS Radical-Scavenging Activity

The antioxidant activities of various solvent extracts were determined by a stable ABTS radical cation, following the method of Re [31], with slight modifications. The generation of the ABTS cationic radical was completed by the reaction of the ABTS solution with potassium persulfate in the dark at room temperature for 12~16 h. In the presence of the hydrogen-supplying antioxidant, the ABTS radical will be reduced. After mixing 7 mM ABTS and 2.45 mM potassium persulfate (final concentration), the resulting ABTS solution was diluted with water so that its absorbance at 734 nm was 0.7 ± 0.05. RA solutions (50 μL) in 95% ethanol at different concentrations (0.5–2.0 mg/mL) were mixed with ABTS solution (1.9 mL), and then the absorbances were read at ambient temperature after 3 min. PBS solution was used as a control. Each sample was operated in parallel three times.

### 4.6. Systematic Understanding of the Mechanism of Action of RA via Computational Target Fishing

#### 4.6.1. Computational Target Fishing by PharmMapper and DRAR-CPI

PharmMapper Server is a freely accessible web-server designed to identify potential target candidates of small molecules for a given probe by means of pharmacogenetic mapping [32]. Meanwhile, the DRAR-CPI server can identify targetable proteins for small molecules by chemical-protein interactome analysis [33]. Both are powerful tools for computational target fishing. The structures of RA were sketched using ChemDraw (Professional 18.0, CambridgeSoft, Cambridge, MA, USA) and uploaded to the PharmMapper and DRAR-CPI servers. All parameters were set to default values, and the top nine protein targets were selected separately from two servers for further investigation.

#### 4.6.2. GO and Pathway Analysis, and Network Construction

To elucidate the function of the targets of RA and its role in signal transduction, the Database for Annotation, Visualization and Integrated Discovery (DAVID, Bioinformatics Resources 6.8, Laboratory of Human Retrovirology and Immunoinformatics (LHRI), Frederick, MD, USA) was used to analyze the GO function and KEGG pathway enrichment of the proteins [34]. The cellular components involved in the target proteins, molecular functions, biological processes and involved pathways were also described. In this research, the network model of the “Compound- Target-Pathway” interaction was established through Cytoscope 3.2.1 (National Institute of General Medical Sciences (NIGMS), Bethesda, MD, USA).

## 5. Conclusions

In summary, a new pyranocoumarin was isolated from ethyl acetate extract of *D. benthami* Prain and was identified as RA. This study was the first to use the SWISS ADME predictor, PASS online servers and the target-fishing method to elucidate the drug likeness, possible biological activity and mechanism of action of RA. The results of the study predict that RA binding shows more abilities to possess better antineoplastic, Topo I inhibitory, anti-angiogenesis and anti-free radical activities, which were reported as the bases of anti-tumor activity. Furthermore, the related in-vitro activities of RA were tested to validate the target fishing results. This serves as the reason why some targets and pathways were associated with cancer, as RA may influence tumor progression through PTK(Syk/Src)-RAS-PI3K/Akt-NFκB. Overall, this study provides a systematic and visual overview of the possible molecular mechanisms and signaling pathways of RA for the treatment of cancer.

## Figures and Tables

**Figure 1 molecules-25-03919-f001:**
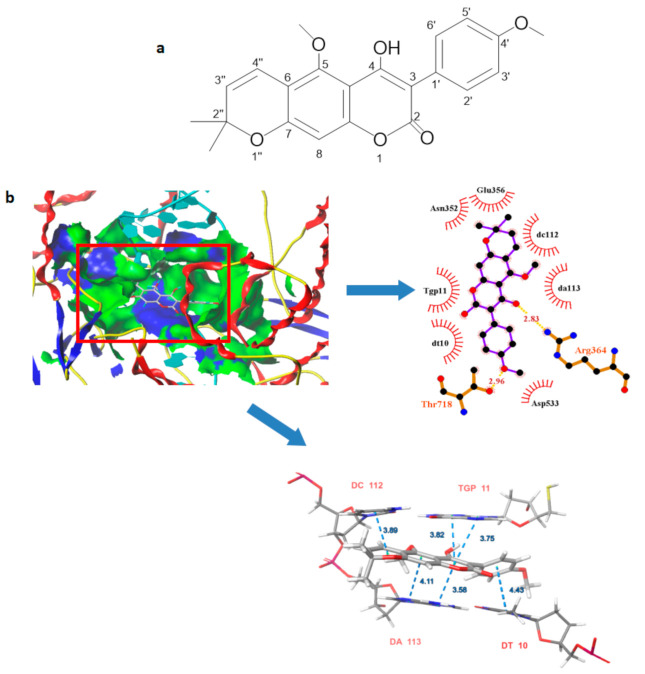
(**a**) The Chemical structure of RA; (**b**) The binding mode of RA with the DNA-Topo I complex (PDB:1T8I).

**Figure 2 molecules-25-03919-f002:**
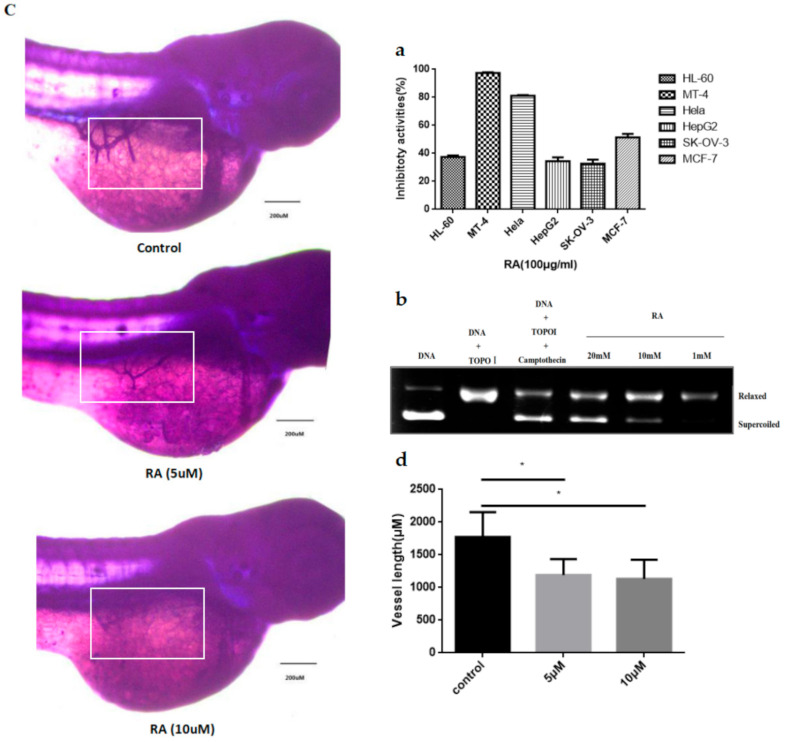
(**a**) The anticancer activities of RA in HL-60, MT4, Hela, HepG2, SK-OV-3and MCF-7 cell lines; (**b**) DNA Topo I inhibitory activity of RA using Camptothecin (0.1 mM) as a positive control; (**c**) Representative photographs of zebrafish SIVs. The SIVs of zebrafish embryos are indicated by a white rectangle; (**d**) Effects of the representative compounds and control on SIVs’ length of 72 hpf zebrafish embryos (x ± s, *n* = 12), * *p* < 0.05; (**e**) DPPH radical scavenging activity; (**f**) ABTS radical scavenging activity.

**Figure 3 molecules-25-03919-f003:**
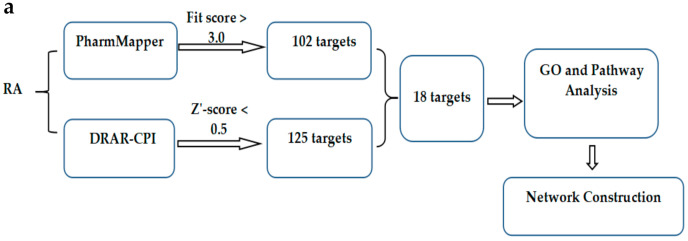
(**a**) The molecular mechanisms of RA via Computational Target Fishing; (**b**) GO analysis of potential targets; (**c**) KEGG pathway analysis of potential targets; (**d**) RA-target-pathway network.

**Table 1 molecules-25-03919-t001:** Pharmacological and molecular properties of RA.

Name	MW (g/mol)	Hdon	Hacc	Rbon	TPSA (Å)	LogP	LogS	Log Kp (cm/s)
RA	380.39	1	6	3	78.13	3.72	−4.80	−5.94

**Table 2 molecules-25-03919-t002:** Activity prediction of RA (only the top ten Pa values are listed).

Pa	Pi	Activity Name
**0.784**	**0.014**	**Antineoplastic**
0.699	0.007	Chemopreventive
**0.655**	**0.005**	**Free radical scavenger**
0.615	0.008	Antiparasitic
0.614	0.012	Antiinfective
0.594	0.009	Antileukemic
0.583	0.004	Antihelmintic
0.570	0.022	Antifungal
0.554	0.042	Antiinflammatory

**Table 3 molecules-25-03919-t003:** The scoring function of ligands and DNA-Topo I.

Compound	Total Score	Crash	Polar	D Score	PMF Score	G Score	CHEM SCORE	C SCORE
Camptothecin	10.33	−0.70	1.15	−176.36	−164.18	−301.14	−13.21	4
RA	7.80	−1.80	0.78	−173.03	−163.80	−249.97	−15.70	4

**Table 4 molecules-25-03919-t004:** Putative targets of RA identified by PharmMapper and DRAR-CPI.

Rank	PDB ID	Name	Target Gene
1	1AXN	Annexin A3	ANXA3
2	1YOL	Proto-oncogene tyrosine-protein kinase Src	SRC
3	2PVY	Fibroblast growth factor receptor 2	FGFR2
4	1J1B	Glycogen synthase kinase-3 beta	GSK3B
5	1JWH	Casein kinase II subunit alpha	CSNK2B
6	1Q11	Tyrosyl-tRNA synthetase, cytoplasmic	YARS
7	1QPC	Proto-oncogene tyrosine-protein kinase LCK	LCK
8	1MQB	Ephrin type-A receptor 2	EPHA2
9	1XBA	Tyrosine-protein kinase SYK	SYK
10	3E92	Mitogen-activated protein kinase 14	MAPK14
11	1S0X	Nuclear receptor ROR-alpha	RORA
12	1CBS	Cellular retinoic acid-binding protein 2	CRABP2
13	1IT6	Serine/threonine-protein phosphatase PP1-gamma catalytic subunit	PPP1CC
14	1BOA	Methionine aminopeptidase 2	METAP2
15	1R1H	Neprilysin	MME
16	1RLB	Transthyretin	TTR
17	3F82	Hepatocyte growth factor receptor	MET
18	3cp9	Vascular endothelial growth factor receptor 2	KDR

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
