# Peer review of "Antitumor Activity and Mechanism of Robustic Acid from Dalbergia benthami Prain via Computational Target Fishing"

_molecules, 2020, doi:10.3390/molecules25173919_

Round 1

Reviewer 1 Report

The manuscript presented by the authors is a nice example of the use of computational tools to propose new bioactivities or potential targets.

I believe it would be of interest of natural products researchers who are trying to include molecular docking and other computational methodologies to their research.

The only correction I suggest is that Figure 1 must be improved as it looks blurry.

Also check the bond length in the structure of RA as most drawing software includes the "clean structure" option.

Good work

Reviewer 2 Report

In this manuscript, robustic acid was extensively studied using various in silico tools. However, it seems that a lot of amendments are needed to be published in Molecules.

Above all, there's no compound called "rustic acid". It seems to have miswritten "robustic acid"

Recently, the authors reported "Molecular Design, Synthesis and Docking Study of Alkyl and Benzyl Derivatives of Robustic Acid as Topoisomerase I Inhibitors" in Chemistry & Biodiversity (2020, vol 17 (3), e1900556). Many important findings of this manuscript are very similar to those in that previous report (such as topoisomerase I inhibitory activity and the molecular docking mechanism of robustic acid to this enzyme, cytotoxicity of robustic acid in HL-60 and HepG2 cell lines).

Although it provides some new information and suggest new target for robustic acid, the most important target is still topoisomerase I as previously reported. 

So I thought the authors should cite the previous report in Chemistry & Biodiversity, minimize the overlapped data and clearly explain what new findings are described in this manuscript compared to the previous one. 

Round 2

Reviewer 2 Report

In the revision statement, the authors answered to the previous comments. However, most of them do not seem to be properly applied to the main text. The authors can compare the results of the present study to those of the previous one in detail. For example, the difference between the conformation selected based on the highest total score and C score.

According to the title, the main finding of this study is the computational target fishing based MOA of robustic acid.  However, there is not much discussion regarding this topic.  

And to me the title is both grammatically and meaningfully a little bit awkward.

Miscellaneously, 

"4.3. Robustic acid isolation" part can be substituted by the reference. 

What is "the half inhibitory activity" in line 98?  

There are many typos. Lower subscripts are not properly used.

There should be space between numbers and units except the temperature unit and %.

There should be also space between words and parenthesis. 
